# Substrate Charge Transfer Induced Ferromagnetism in MnSe/SrTiO_3_ Ultrathin Films

**DOI:** 10.3390/nano14161355

**Published:** 2024-08-16

**Authors:** Chun-Hao Huang, Chandra Shekar Gantepogu, Peng-Jen Chen, Ting-Hsuan Wu, Wei-Rein Liu, Kung-Hsuan Lin, Chi-Liang Chen, Ting-Kuo Lee, Ming-Jye Wang, Maw-Kuen Wu

**Affiliations:** 1Institute of Physics, Academia Sinica, Taipei 11529, Taiwanwthse0423@gate.sinica.edu.tw (T.-H.W.); linkh@phys.sinica.edu.tw (K.-H.L.); mkwu@phys.sinica.edu.tw (M.-K.W.); 2Department of Physics, National Taiwan University, Taipei 10617, Taiwan; 3Nano Science and Technology Program, Taiwan International Graduate Program, Academia Sinica, Taipei 11529, Taiwan; 4Physics Division, National Center for Theoretical Sciences, Hsinchu 30013, Taiwan; 5National Synchrotron Radiation Research Center, Hsinchu 30076, Taiwan; 6Department of Physics, National Sun Yat-sen University, Kaohsiung 80424, Taiwan; 7Institute of Astronomy and Astrophysics, Academia Sinica, Taipei 10617, Taiwan

**Keywords:** MnSe, ultrathin film, tetragonal, giant conductivity enhancement

## Abstract

The observation of superconductivity in MnSe at 12 GPa motivated us to investigate whether superconductivity could be induced in MnSe at ambient conditions. A strain-induced structural change in the ultrathin film could be one route to the emergence of superconductivity. In this report, we present the physical property of MnSe ultrathin films, which become tetragonal (stretched *ab*-plane and shortened *c*-axis) on a (001) SrTiO_3_ (STO) substrate, prepared by the pulsed laser deposition (PLD) method. The physical properties of the tetragonal MnSe ultrathin films exhibit very different characteristics from those of the thick films and polycrystalline samples. The tetragonal MnSe films show substantial conductivity enhancement, which could be associated with the presence of superparamagnetism. The optical absorption data indicate that the electron transition through the indirect bandgap to the conduction band is significantly enhanced in tetragonal MnSe. Furthermore, the X-ray Mn *L*-edge absorption results also reveal an increase in unoccupied state valance bands. This theoretical study suggests that charge transfer from the substrate plays an important role in conductivity enhancement and the emergence of a ferromagnetic order that leads to superparamagnetism.

## 1. Introduction

Transition metal compounds (TMCs) have attracted much attention in recent decades because of their wide range of electronic behaviors and the possibility to observe superconductivity, as discovered in iron-based chalcogenides and pnictides [1,2,3,4]. However, transition metal compounds often exhibit magnetism that competes with superconductivity. Modifying the crystal structure–property correlation is thus essential to fine-tune the interplay between magnetism and superconductivity in TMCs. It has been demonstrated that chemical substitution or external pressure can modify the metal–ligand bond distance and the nearest neighbor environment of TMC, which results in different electric and magnetic properties. For example, the induced internal pressure in BaFe_2_As_2_ by substituting some potassium for barium suppresses the tetragonal (I4/mmm) to orthorhombic (Fmmm) phase transition at ambient conditions. The subsequent reductions in As-Fe-As bond angle and Fe-Fe distance were considered the source for the emergence of superconductivity in a BaFe_2_As_2_ system [3,4]. Superconductivity was also observed in CrAs and MnP after their antiferromagnetism [5,6,7] and helimagnetism [7,8] was suppressed, respectively, under hydrostatic pressure [9,10]. The NaCl-type MnSe exhibits a complex magnetic property [11]. Strong thermal hysteresis and magnetic anomalies between 100 K and 200 K were observed [12,13]. Our early results showed that the observed magnetic anomaly originated from the partial structural transition of NaCl-type MnSe to the hexagonal phase at a low temperature [14]. Recently, a superconducting transition near 5 K, which is significantly higher than that of MnP, was observed in MnSe at pressures above 12 GPa [15]. The superconducting phase is closely related to the appearance of an orthorhombic structure, which distorts the Mn-ligand-center octahedrons. Furthermore, a recent study demonstrated that the changes in crystal structure and electrical property in Ti_2_O_3_ films are due to the strain effect at the substrate–film interface [16]. These results inspired us to investigate the possibility of inducing superconductivity in a MnSe film at ambient pressure through epitaxial strain.

MnSe films have been prepared by different techniques, such as molecular beam epitaxy (MBE) [17,18,19,20], organometallic vapor phase epitaxy [21], thermal evaporation [22], the potentiostatic cathodic electrodeposition technique [23], and chemical bath deposition [24]. Most of the reported MnSe results are polycrystalline films. Here, we report MnSe films with highly preferred orientations deposited via the pulsed laser deposition (PLD) method on an STO substrate. The in-plane lattice constant is slightly larger than the *c*-axis in the MnSe ultrathin films, indicating that the crystal symmetry was transformed from cubic to tetragonal. The tetragonal MnSe films reveal a substantial conductivity enhancement, and the results could be associated with the existence of superparamagnetism induced by the charge transfer from the STO substrate. The optical and X-ray Mn *L*-edge absorption results provide evidence for the proposed scenario of physical property changes on strained MnSe ultrathin films.

## 2. Materials and Methods

A single-phase polycrystalline MnSe target was prepared using the solid-state reaction method. The high-purity Mn (99.95%, Alfa-Aesar, Haverhill, MA, USA) and Se (99.95%, Acros-Organic, Geel, Belgium) powders were mixed in stoichiometric proportions and thoroughly ground. The mixture was sealed in an evacuated quartz tube with a pressure near 10^−3^ bar. Then, the mixture was sintered at 750 °C in a furnace for several hours. The sintered mixture was subsequently ground and sintered again (sealed in an evacuated quartz tube) at 750 °C for several hours to form a uniform single-phase polycrystalline MnSe power. The powder was pressed into a one inch circular disk with a thickness of a few millimeters. Finally, the disk was sintered at 400 °C in an evacuated sealed quartz tube to form a solid MnSe target.

The MnSe films were deposited onto a (001) STO substrate via the PLD (Lambda Physik KrF (λ = 248 nm)) method with a laser repetition rate of 5 Hz. All depositions were carried out in a vacuum chamber with a base pressure of 7 × 10^−7^ Torr and a substrate temperature of 600 °C. The substrate-to-target distance was about 5 cm. The thickness of the MnSe film was roughly controlled by the deposition time and precisely determined from the cross-sectional TEM (transmission electron microscopy) image, as shown in Figure 1. The films for the TEM measurement and other studies are actually analogous samples. The measured thicknesses were used to calculate the electrical resistivity and magnetic susceptibility.

The physical properties of the selected samples with varying thicknesses were studied. The structures of the deposited films were characterized by the four-circle diffractometers at the TLS 13 and TPS 09 beamlines of the National Synchrotron Radiation Research Center (NSRRC), Taiwan. The resistivity of films was measured by standard four-terminal resistance measurement in a PPMS (Physical Property Measurement System, QUANTUM DESIGN). The magnetic susceptibility measurement was performed by a QUANTUM DESIGN metal shielded 7-T VSM SQUID (superconducting quantum interference device) magnetometer. The magnetic susceptibility contributed by the STO substrate was removed from the bare substrate measurement data. The optical transmission spectra of the MnSe films and bare STO substrate were measured with the light source of a halogen–tungsten lamp. The transmittance of the MnSe films was extracted from the ratio between the spectra of the films and the bare substrate. The Mn L-edge X-ray absorption near edge spectrum (XANES) was measured in the total electron yield (TEY) mode at the TLS BL20A beamline of NSRRC Taiwan with resolving power E/ΔE = 8000 in an ultra-high vacuum chamber (<5 × 10^−9^ torr). Standard oxides were used for energy calibration. All spectra were normalized by a standard procedure. The density functional theory (DFT) calculations were performed using QUANTUM ESPRESSO [25] with a k-grid. Ultrasoft Perdew-Burke-Ernzerhof (PBE) functionals were used to account for the exchange-correlation effect and the energy cutoff is chosen to be 50 Ry.

## 3. Results

Figure 2 displays the temperature-dependent resistivity of the MnSe films. The resistivity of all investigated films shows semiconductor-like characteristics. The thermal hysteresis loop (near room temperature) of resistivity reported in polycrystalline MnSe [26], attributed to the partial phase transformation from cubic to hexagonal [11,12], was not observed in all film samples. The resistivity of the films at room temperature is apparently smaller than the values reported for the bulk sample [26,27] and the polycrystalline film [28]. In other words, the conductivity of the films is clearly enhanced. To further investigate the reason for the conductivity enhancement, the absorption spectra of the films were measured, as shown in Appendix A. According to the Bouguer–Lambert–Beer absorption law, the optical transmittance of a material follows the equation *T* = *e*^−*αd*^, where *α* is the absorption coefficient and *d* is the thickness of the sample. The optical direct and indirect bandgap energies in the MnSe films are determined by extrapolating the linear least squares fit using *α*^2^-*E* and *α*^1/2^-*E* plots, respectively. The transmittance data indicate that the 740 nm film has a direct bandgap of 2.6 eV and the 30 nm film has an indirect bandgap of 0.63 eV, as shown in Appendix A. The notable decrease in the electron transition bandgap from the valence band to the conduction band for the 30 nm film leads to a greater presence of thermally excited electrons in the conduction band. This phenomenon elucidates the substantial enhancement observed in its conductivity compared with the 740 nm film.

The M-T curves of the investigated films as well as polycrystaline MnSe are shown in Figure 3. The magnetizations of the 740 nm and 140 nm films are much smaller in comparison with that of the 30 nm film. The M-H curves of the samples at 300 K, as shown in Figure 4a, exhibit a rapid increase at the low magnetic field region and saturation at the high magnetic region. Moreover, the M-H curves of the 20 nm film in the low magnetic field region, as shown Appendix A, reveal a very small hysteresis loop (or coercive field). We also conducted M-H measurements over wider temperature ranges for the 40 nm MnSe film, as depicted in Appendix A. These observations can be well described by the superparamagnetic characteristics in these MnSe films. Additionally, the thickness-dependent behavior of magnetic susceptibility strongly suggests that the superparamagnetism most likely exists near the interface with the substrate.

Typically, the superparamagnetic materials consist of nanometer-scale magnetic domains. Assuming that the individual domain size and the magnetic moment are identical, the magnetization of the superparamagnetic material at a high temperature follows M = n*μL*(*μ*_0_*μH/k_B_T*), where *n* is the density of the superparamagnetic domain, *μ* is the magnetic moment of each superparamagnetic domain, *μ*_0_ is the magnetic permeability of free space, *k_B_* is the Boltzmann constant, *T* is the temperature of the material, and *L* is the Langevin function. The values of *n* and *μ* can be extracted by fitting their M-H curve at high temperatures. Figure 4b demonstrates excellent fitting of the calculated M-H curve (red line) to the measured data (open circles). The estimated density of the magnetic domain is 5.74 × 10^18^ cm^−3^, corresponding to a domain size of 5.58 nm. The extracted magnetic moment of the individual domain is 13,786 *μ_B_* and the effective moment of Mn is 3.21 *μ_B_*, where *μ_B_* is the Bohr magneton (~9.274 × 10^−21^ emu).

Another important signature of superparamagnetism is the blocking temperature, which is related to the barrier energy from moving in the direction of magnetization. Below the blocking temperature, the magnetization decreases significantly because of a freeze in the magnetic moment. The 7 nm MnSe nanoparticle shows a blocking temperature near 7 K [29]. However, this blocking temperature is not observed in the M-T curves of the 30 nm film above 2 K, as shown in Figure 3d, indicating that the energy barrier is low due to the small domain size (5.58 nm).

To gain more insight into the observed enhancements in both conductivity and magnetic susceptibility in the MnSe thin film samples, we conducted detailed analyses of the high-resolution X-ray diffraction results. The X-ray radial scans along the surface normal of the MnSe films with thicknesses of 30, 140, and 740 nm are shown in Figure 5a. Besides the peaks at q ~ 1.609 and 3.219 Å^−1^ from the STO substrate, the peaks at q ~ 2.3 Å^−1^ and 4.6 Å^−1^ are from the cubic MnSe phase, with Miller indices of (002) and (004). The X-ray radial scan results indicate that the MnSe films are well grown along the c-axis. Figure 5b shows the diffraction pattern near the MnSe (002) peak. The (002) peak of the 30 nm film is shifted to a high angle, which suggests a shorter c-axis lattice constant. The azimuthal φ scan of the (022)_MnSe_ and (101)_STO_ peaks, as shown in Figure 5c, demonstrate a good 4-fold symmetry of the 30 nm MnSe film with a 45° rotation with respect to the (101)_STO_ peaks, indicating that the *a*- (or *b*-) axis of MnSe film oriented preferentially grows along the (110) direction of STO to match the substrate lattice.

The in-plane lattice parameters of the MnSe films were obtained from the (200) and (020) peaks of the grazing incidence X-ray diffraction. Figure 5d,e show the diffraction profiles of MnSe films with thicknesses of 140 nm and 30 nm, respectively. For the 140 nm film, the peak positions of (200), (020), and (200) are close, correspondingly having similar lattice constants along the *a* (5.4561 Å), *b* (5.4557 Å), and *c* (5.4620 Å) axes. These values are similar to the lattice constant (5.45 Å) of the polycrystalline cubic MnSe [11,14]. On the contrary, the (200) and (020) peaks of the 30 nm film, as shown in Figure 5e, appear almost at the same angle, but the (002) peak is located at a slightly higher angle. The calculated lattice constants of *a* and *b* are nearly identical, 5.4843 Å and 5.4835 Å, respectively, while the lattice constant *c* is approximately 0.7% smaller, at 5.4472 Å.

The extended lattice constant of STO along the (110) direction is 5.5225 Å ($\sqrt{2}$
*a*_STO_ = $\sqrt{2}$ × 3.905), which is larger than the bulk MnSe of 5.45 Å. The in-plane lattice of a MnSe film in contact with the STO substrate experiences a tensile strain, resulting in an enlarged *ab*-plane and a shortened *c*-axis. The crystal symmetry changes from cubic to tetragonal with an averaged *c*/*a* ratio of 0.9938. As the film thickness increases, as in the 140 nm film’s case, the tensile strain on the upper layer of the film (>few tens of nanometers) relaxes gradually, and the cubic crystal symmetry is restored, which dominates the X-ray measurement results.

Now, we can clearly correlate the observed resistivity and magnetic anomalous behavior with the appearance of a tetragonal phase near the film-substrate interface. The emergence of the tetragonal phase in 30 nm MnSe provides a new channel for excited electrons to transition from the valence band to the conduction band, consequently leading to a decrease in resistivity. The calculated magnetic susceptibility is a volume average of the stretched (tetragonal) and non-stretched (cubic) MnSe layers in the film. As the film thickness increases, the volume ratio of non-stretched cubic MnSe increases, resulting in a decrease in the calculated magnetic susceptibility. Similarly, the stretched and non-stretched layers also contribute to the calculated resistivity. The resistivity can be dominated by the stretched MnSe layer if its resistivity is much smaller than that of the non-stretched layer. For example, the 740 nm film exhibits a bulk-like X-ray diffraction characteristic, while its electrical resistivity is around 0.1–0.2 kΩ-cm at 300 K, which is much smaller than that of the polycrystalline MnSe sample, 1–10 kΩ-cm, reported by Takashi Ito et al. [27].

A further investigation into the strain effect was carried out by the XANES in the TEY mode. The full Mn L-edge XANES of the studied films are shown in the inset of Figure 6. The spectra exhibit several well-resolved features, showing that 2*p* core electrons are excited to Mn 3*d* unoccupied states. These spectra arise from the photon–electron transition from the 2*p*^6^3*d*^n^ ground state to the 2*p*^5^3*d*^n+1^ excited state, involving multiple excitations in the energy range 635–646 eV and 648–656 eV, noted as L_3_ (2*p*_3/2_→3*d*) and L_2_ (2*p*_1/2_→3*d*) absorptions, respectively [30]. The Mn L_3_-edge XANES spectra, shown in the main frame of Figure 6, exhibit a multi-peak structure under O*_h_* crystal field splitting, i.e., the 3*d* unoccupied state splits into t_2g_ and e_g_ sub-bands due to spin-orbit coupling. The spectra are consistent with an earlier report [31], indicating that Mn ions in all MnSe films are in the Mn^2+^ state. Three main absorption peaks can be easily identified, marked as B, C, and D. Peak A is not obvious because of overlaps with peak B. After the continuous absorption background with an arctangent functional form was subtracted, these spectra were fitted with five Gaussian peaks. The fitting results and parameters of each peak are presented in Appendix A. The parameters of the relevant peaks (A, B, C, and D) are tabulated in Appendix A.

Compared with the 740 nm film, each peak (of the 140 nm and 30 nm films) has different variations when the *ab*-plane of MnSe is stretched. Peak A has a significant increase in area and FWHM. Peak B also has an obvious increase in area but with the same FWHM. Peaks C and D clearly become narrower (a smaller FWHM). However, the area of peak C increases, while that of peak D decreases. Peak B is associated with the unoccupied states of the Mn t_2g_ band, which is the closest band to the Fermi level (E_F_) and not hybridized with the Se 4*p*^2^ states. The increase in area indicates an increase in the available states in the conduction band. As discussed earlier, the optical absorption result shows that the MnSe films are semiconductors with a direct gap of 2.6 eV and an indirect bandgap of 0.7–1.0 eV. From Appendix A, the energy difference between peaks A and B is 0.706, 0.77, and 0.72 eV for the 740 nm, 140 nm, and 30 nm MnSe films, respectively, which is consistent with the values of the indirect gap energy. Therefore, peak A can be attributed to the absorption of the unoccupied (hole) states in the valance band, which are the e_g↑_-4*p*^2^_↑_and e_g↓_-4*p*^2^_↓_ hybridized states [32]. The area of peak A (or the number of holes in the valance band) increased in the 30 nm film, revealing that the transition of an electron from the valance band to the indirect band was enhanced. This result is consistent with the conclusion of the optical absorption experiments and the enhancement of conductivity in the electrical transport measurement. Peaks C and D are associated with the states of the 3*d*^4^L character, where L means a hole in the ligand. These hybridized states are sensitive to the Mn-Se lattice structural symmetry of MnSe. For example, the states next to t_2-_ are the hybridized states between the Mn e_g↓_ and the Se 4*p*^2^_↓_, marked as *AB_−_* in [32]. The e_g↓_ states consist of 3*d*_x2-y2↓_ and 3*d*_3z2-r2↓_ orbitals. For the 140 nm and 30 nm MnSe films, the in-plane Mn-Se distance was elongated and the out-of-plane Mn-Se distance was shortened. The polarization orbital 3*d*_x2-y2↓_-4*p*_x,y↓_ with the *a*- and *b*-axis coupling of spin-down would weaken the hybridized orbital, while the 3*d*_3z2−r2↓_-4*p*_z↓_ with the characteristic *c*-axis orbital would be enhanced. Consequently, the Mn3*d*-Se2*p* orbital hybridized band changes in the conduction band are mainly due to the influence of the substrate stress on the MnSe lattice structure, which leads to the modulation of O*_h_* symmetry.

As there is no complete experimental evidence to demonstrate the origin of superparamagnetism to explain our observation, we have carried out density functional theory (DFT) calculations to find out whether there is any hint about the source of superparamagnetism. The calculations used the ultrasoft Perdew–Burke–Ernzerhof (PBE) functionals to account for the exchange correlation effect with an energy cutoff of 50 Ry. A tetragonal unit cell was used for both the A-AFM (A-type antiferromagnetism) and FM (ferromagnetism) configurations (here, neglecting all interface and surface effects). With (001) termination, the A-AFM phase is a magnetic ground state (near the interface) and the FM phase is 10.6 meV (per formula unit) higher than the A-AFM phase. Figure 7 shows the bulk band structures of A-AFM and FM MnSe. Considering the underestimation of the bulk band gap of 0.7 eV [33], a rigid shift of 0.7 eV is applied to the conduction bands of both phases to approximately correct the band structure. It can be seen that both the A-AFM and FM phases are semiconducting with an indirect gap roughly estimated to be 0.7 eV (although a more accurate band gap can be obtained by using hybrid functionals [34,35] or GW approximation [36,37], it is not the main focus of this work and thus only the results using PBE functionals are demonstrated). Consistent with the absorption spectra observed, the gap reduces in magnitude and becomes indirect.

As revealed in the Mn L-edge XANES results, the number of holes increases as the MnSe film becomes thinner. Consistently, the Ti L3-edge absorption spectrum of the STO substrate with the 30 nm MnSe film shows a clear reduction in the unoccupied states, included in Appendix A, indicating that the STO substrate plays the role of a hole donor to the MnSe near the interface. Indeed, the Seebeck measurements, as shown in Appendix A, clearly show the presence of a p-type carrier for the 30 nm and 730 nm films. The DFT calculations indicate that the FM phase becomes energetically favored when the (homogeneous) concentration of holes is higher than 5 × 10^21^ cm^−3^. The much smaller hole concentration, 2.07 × 10^18^ cm^−3^, estimated from the Seebeck measurement data, in comparison with the theoretical predicted critical value, is most likely due to the film measured, which has a thickness much larger than the domain-size estimated from the magnetic measurements.

The charge transfer phenomenon at the interface of two perovskite ABO_3_ insulators, such as SrTiO_3_, has been widely studied. Zhong et al. proposed a scheme to predict the sign and relative amplitude of the intrinsic charge transfer [38]. In addition, the A site cation could be crucial for controlling the interfacial charge transfer direction [39]. Although MnSe is not a perovskite insulator, the atomic arrangement along the *c*-axis of the MnSe (100) on SrTiO_3_ (110), displayed in Appendix A, is similar to that near the interface of the ABO_3_/AB’O_3_ perovskites, as shown in Figure 1b of reference [38]. The electron of Mn could transfer to the Ti^4+^ site of the SrTiO_3_ due to the same charge transfer mechanism. We believe that the MnSe ultrathin film on other similar perovskite (ABO_3_) insulating substrates might have a similar charge transfer phenomenon.

If FM domains, rather than the ordered FM phase, are formed, the holes tend to accumulate within these domains and the required high hole concentration may be locally achieved. Together with the energy gain from the Zeeman effect when an external magnetic field is turned on, the formation of FM domains can be energetically favored. The presence of doped domains can also explain the thickness dependence of the observed conductivity. Because the holes are confined within the FM domains that are separated by the insulating environment (the A-AFM region in the bulk), the separation between these conducting domains becomes larger when the film becomes thicker, which in turn decreases the conductivity.

## 4. Conclusions

We have successfully prepared MnSe thin films with a highly preferred orientation on an STO substrate using PLD. The lattice mismatch from the substrate induces a strain on the MnSe film near the interface and, according to the X-ray diffraction data, results in a transformation from cubic to tetragonal crystal symmetry for the MnSe ultrathin films. Though there is no indication of the emergence of superconductivity, the tetragonal MnSe ultrathin films show a very different physical property from the thick film or bulk sample. The tetragonal 20 nm MnSe film indicates superparamagnetism at room temperature with a magnetic domain size of 5.58 nm estimated from the M-H curve. Due to the small domain size, the magnetic (blocking) temperature is not found at a temperature higher than 2 K. The films also reveal a large conductivity enhancement. The optical absorption and X-ray Mn L-edge absorption results suggest that the electron transition to the indirect conduction band is significantly promoted. The stretched *ab*-plane and shortened *c*-axis would modify the coupling between the Mn 3*d* and Se 4*p* polarization orbitals. The theoretical (DFT calculation) and XANES studies imply that the charge (hole) transfer from the STO substrate makes the ferromagnetic phase of MnSe energetically favored and results in the emergence of superparamagnetism due to the substrate charge transfer inducing ferromagnetic order. Further investigations of thinner films, even down to a monolayer, will be important to confirm our claims and to examine superconductivity in MnSe at ambient conditions.

## Figures and Tables

**Figure 1 nanomaterials-14-01355-f001:**
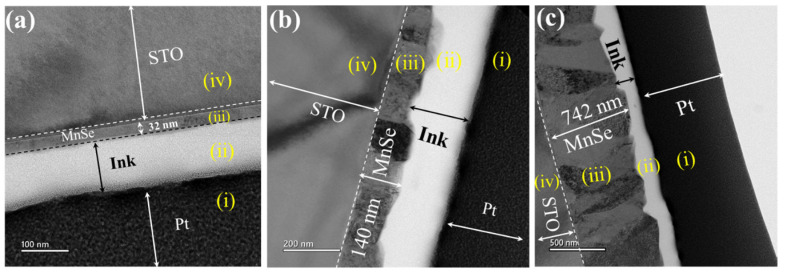
Cross-sectional TEM image of the three studied MnSe films. The measured thicknesses are (**a**) 30 nm, (**b**) 140 nm, and (**c**) 740 nm. The ink (white layer in each image) is a carbon paste used as a layer on the top of the films to protect from damage during the TEM sample preparation process. The regions (i), (ii), (iii), and (iv) marked in the images are Pt (conduction layer for FIB process), ink (organic protection layer), MnSe film, and STO substrate.

**Figure 2 nanomaterials-14-01355-f002:**
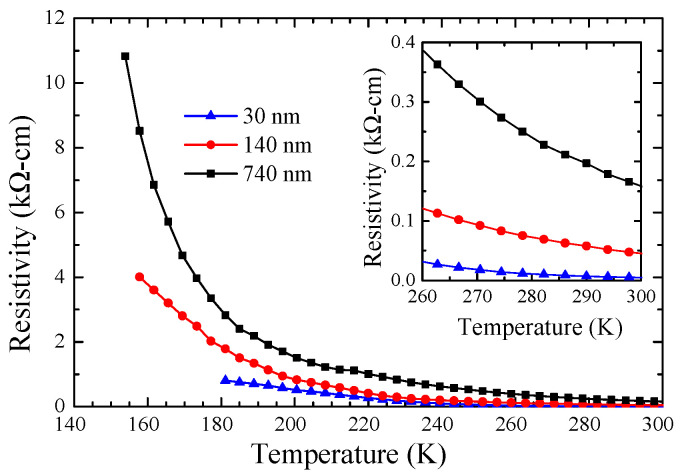
The temperature dependence of the resistivity of MnSe films. The resistivity decreases dramatically as the thickness of the film becomes thinner, more than one order of magnitude smaller in the 30 nm film than that of the 740 nm film.

**Figure 3 nanomaterials-14-01355-f003:**
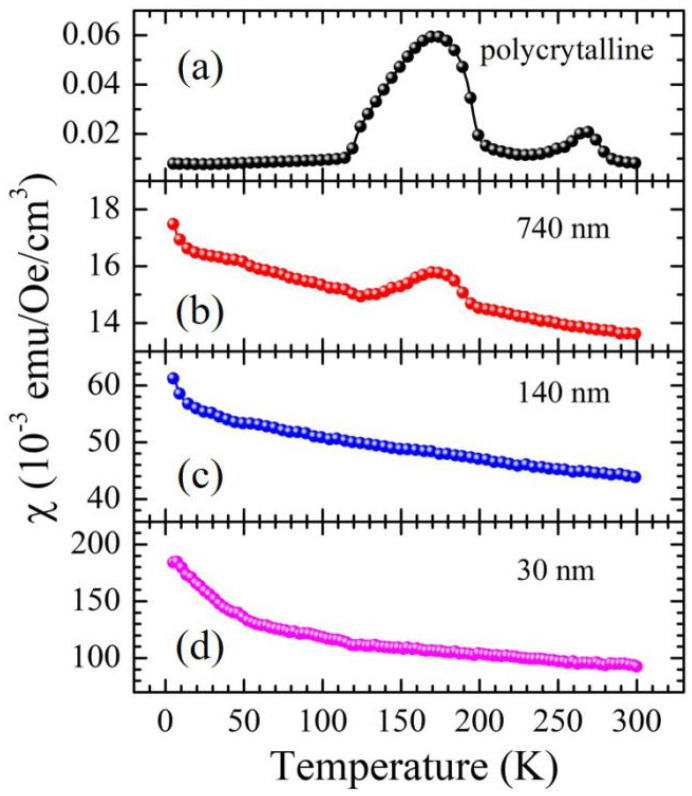
The magnetic susceptibility of (**a**) polycrystalline MnSe powder and the (**b**) 740 nm, (**c**) 140 nm, and (**d**) 30 nm MnSe films. The polycrystalline MnSe sample was prepared via the solid-state reaction method. Its magnetic characteristics exhibit anomalies at 180 K and 270 K, respectively. The magnetic anomaly at 270 K arises from the antiferromagnetic order of the solitary hexagonal phase, with a partial transformation into the cubic phase [14]. The magnetic anomaly observed at 180 K is attributed to the antiferromagnetic order of both the collective hexagonal and cubic phases [14]. The 180 K magnetic anomaly in the polycrystalline sample is only observed in thick (740 nm) films. It is noted that the magnetic susceptibility of the film increases substantially as film thickness decreases.

**Figure 4 nanomaterials-14-01355-f004:**
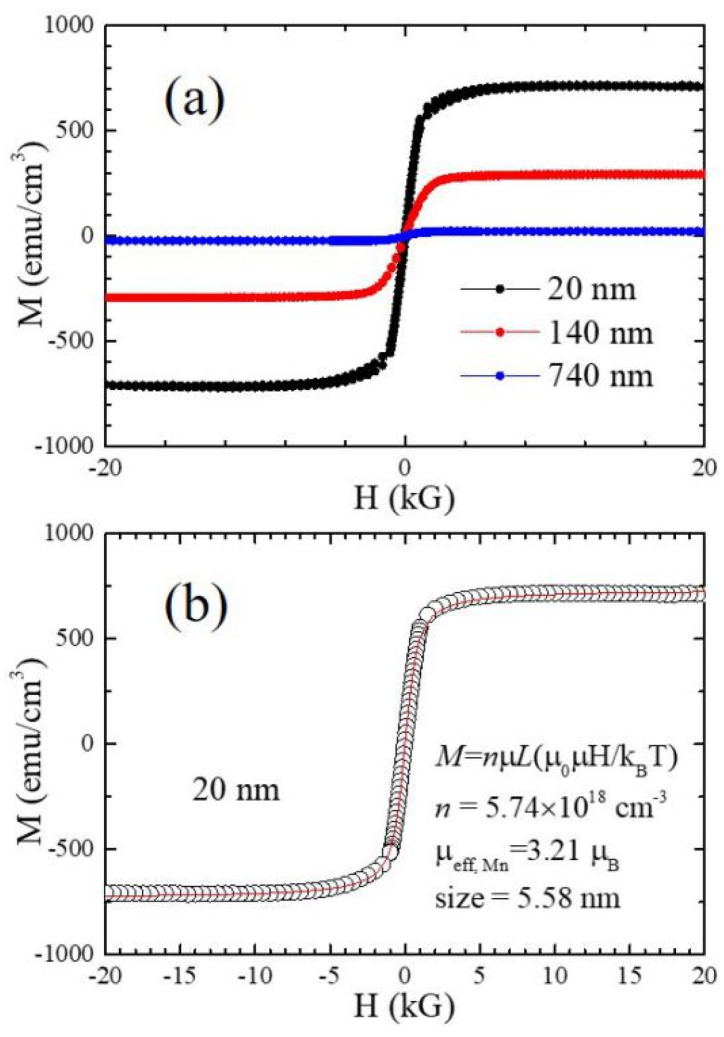
An M-H curve of the MnSe films. (**a**) The M-H curve of films reveals the signatures of superparamagnetism, rapid increase at the low magnetic field region, saturation at the high magnetic region, and no hysteresis (zero coercive field). There is a small hysteresis loop in the low field region, as shown in Appendix A. The moment is in the order of 10^−4^ emu and higher, which is well beyond the sensitivity of SQUID, ~10^−6^ emu. (**b**) The fitting of the M-H curve of the 20 nm film. The fitting curve (red line) agrees excellently with the experimental data (open circles). The extracted density of the magnetic domain is about 5.74 × 10^18^ cm^−3^, and the effective magnetic moment of Mn is about 3.21.

**Figure 5 nanomaterials-14-01355-f005:**
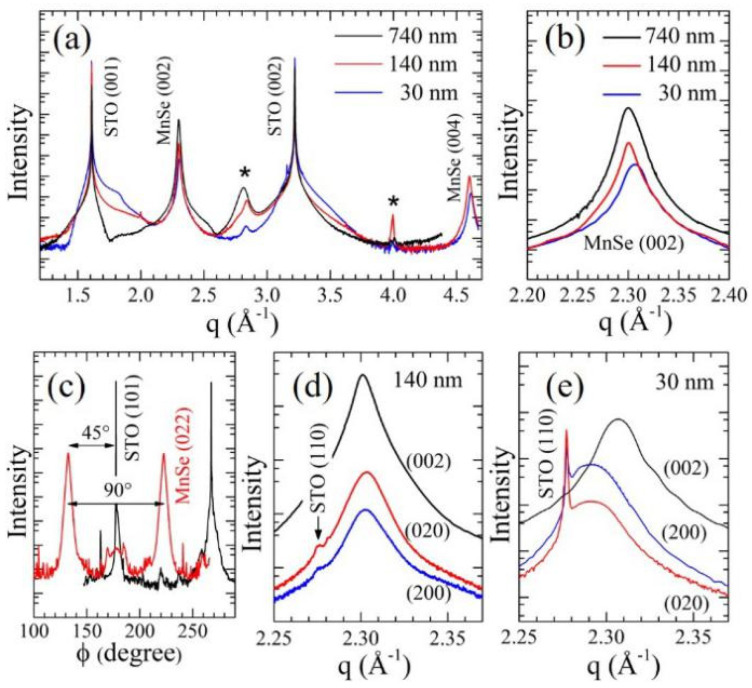
X-ray diffractions of the MnSe thin films. (**a**) A radial scan of the MnSe films with thicknesses of 30 nm, 140 nm, and 740 nm on the (001) STO substrate. The films were grown with the c-axis preferred orientation. The * mark (>3 orders of magnitude smaller than the (001) peak) at q near 2.8 Å^−1^ and 4.0 Å^−1^ are indexed as (112) and (222) peaks of MnSe. (**b**) A radical scan near the MnSe (002) peak. The peak of the 30 nm film shifts to a higher angle, implying a shorter *c*-axis lattice constant. (**c**) ψ-scan profiles with respect to the (022)_STO_ and (101)_MnSe_ diffraction peaks of the 30 nm film. A 4-fold symmetry in the *ab*-plane at 45° with respect to the STO *a*- (*b*-) axis demonstrates the epitaxial growth of the MnSe film. (**d**) The (200), (020), and (002) diffraction peaks of the 140 nm film. Their positions have no significant difference. (**e**) The (200), (020), and (002) diffraction peaks of the 30 nm film. The in-plane diffraction peak is located at a lower angle compared with the (002) peak.

**Figure 6 nanomaterials-14-01355-f006:**
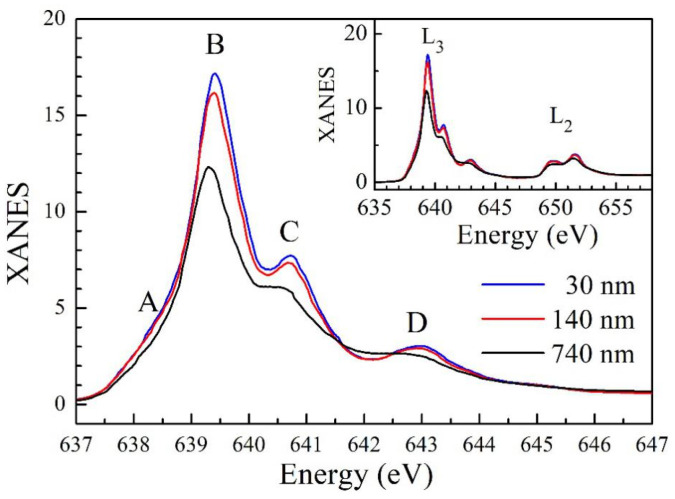
Mn L-edge X-ray absorption spectra of the studied MnSe films. The XANES (X-ray absorption near edge spectroscopy) spectra of the Mn 2*p*_3/2_ to 3*d* (L_3_) transition. Four peaks are identified, (marked as A, B, C, and D) which originated from different Mn 3*d*-related final states. The inset shows the full spectrum of L-edge absorption.

**Figure 7 nanomaterials-14-01355-f007:**
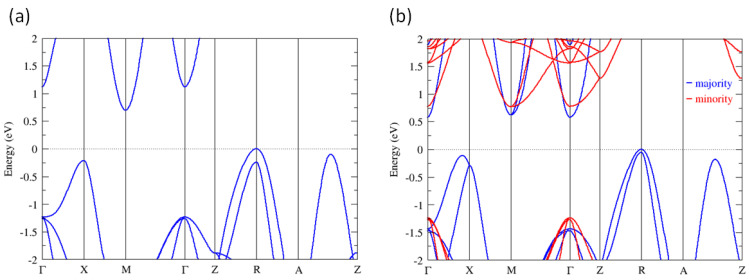
The band structure of tetragonal MnSe. (**a**) The A-AFM phase. (**b**) The FM phase. For comparison, an identical unit cell is used for both phases. A rigid shift of 0.7 eV is applied to the conduction bands, as mentioned in the text.

## Data Availability

All data supporting the findings of this work are available from the corresponding author on request.

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
