# Peer review of "Substrate Charge Transfer Induced Ferromagnetism in MnSe/SrTiO3 Ultrathin Films"

_nanomaterials, 2024, doi:10.3390/nano14161355_

Round 1

Reviewer 1 Report (New Reviewer)

Comments and Suggestions for Authors

The paper is deserved to be published. I have only few questions/suggestions to authors:

Why the parts of text are colored?

Can you present HRTEM of the interface?

Why did you choose 30nm, 140nm and 740 nm?

Where is the third thickness at Figure S1?

Author Response

Reviewer 2 Report (New Reviewer)

Comments and Suggestions for Authors

The manuscript investigates substrate charge transfer-induced ferromagnetism in MnSn ultrathin films deposited by PLD. The conclusions are supported by a variety of experimental methods and theoretical calculations. These results are novel and interesting. However, before considering the manuscript for publication, the authors should undertake revisions aimed at reorganizing the manuscript and providing additional experimental details.

1) Due to the high temperature of the STO substrate during MnSe deposition in the PLD chamber, the substrate might outgas, inducing oxygen vacancies. What are the magnetic properties of the STO substrate? Have the M-H and M-T data obtained from the SQUID been corrected for the substrate background signal? The authors should provide data for the annealed STO substrate for comparison.

2) Additionally, what about the effects of other substrates on the magnetic properties of the MnSn ultrathin film? Can this charge transfer-induced ferromagnetism be found in the other substrates? The authors should include a discussion on this topic.

3) Since the theoretical results strongly support the main conclusion of this manuscript, these calculated results should be included in the main text.

Round 2

Reviewer 2 Report (New Reviewer)

Comments and Suggestions for Authors

This manuscript is suitable for acceptance in its current form.

This manuscript is a resubmission of an earlier submission. The following is a list of the peer review reports and author responses from that submission.

Round 1

Reviewer 1 Report

Comments and Suggestions for Authors

referee report 

nanomaterials-2941860-peer-review-v1

Substrate charge transfer induced ferromagnetism in MnSe/SrTiO3 ultrathin films 

Chun-Hao Huang et al.

The present manuscript describes the preparation of MnSe/SrTiO3 ultrathin films and their characteriation, stimulated by search

for superconductivity at ambient conditions. Instead, the obtained samples exhibit paramagnetic behavior. The topic is interesting 

and fits well to the scope of Nanomaterials.

The manuscript comprises 6 figures, no table, a supplementary material with several more figures and infos, and 39 references are given.

The style of the referenes does not follow the journal guidelines.

On the first glance, this is a well written manuscript. However, careful reading reveals some problems, enlisted below:

# Too much impoerant information ended up in the supplemental material, which is painfully missing in the manuscript. Please add Sec. 1

  to the main text -- this will help a lot. Further, the description of the resistance measurement is only minimal -- there should be

  information concerning the measurement units, the contact preparation, the arrangent of the contacts and the distance of the voltage

  pads, and finally, the currents applied. Also the TEM images would make a good part within the main text.

Some minor points:

# There should always be a space between a physical quantity and its unit

# Please make sure to define all abbreviations at their first appearance

Overall, the data provided are very interesting, but the points mentioned above should be considered. Thus, the manuscript should 

undergo a major revision.  

Comments on the Quality of English Language

Language is fine, some coorections required

Reviewer 2 Report

Comments and Suggestions for Authors

The paper by Huang et al. reports on induced ferromagnetism in MnSe films epitaxially grown on SrTiO3. Authors attribute the increased magnetic susceptibility and conductivity to the charge transfer from the substrate as well as a tetragonal distortion of the MnSe layers. They used multiple techniques such as resistivity and Seebeck coefficient measurements, magnetometry measurements (susceptibility and magnetization), TEM, XRD, optical absorption, XPS, XANES and theoretical calculations to conclude about effect of strain which might be of potential interest for further step towards room-temperature superconductivity.

The paper presents an interesting experimental results on strained and unstrained (thick) MnSe films but lacks the sufficient explanation and therefore has to be revised before the next consideration as it contains several unclear points and inconsistencies summarized below. My main concern addresses the discussion of magnetic properties - Authors claim they have superparamagnetism in a film (not in clusters, etc) which is highly doubtful, in my opinion, they observed a ferromagnet with low coercivity. But here another problem comes - the value of magnetization seems to be surprisingly (unrealistic) high (750 emu/cm3 or 750 kA/m) and requires further clarifications on how it was obtained (what is the raw data on magnetic moment of all measured films?) and how carefully external sources of contamination were excluded.

I suggest to consider the following questions (numbered sequentially, not accordingly to the rank of importance):

1. I think that readers will appreciate more detailed description of PLD target synthesis (in Suppl.) - Authors write about sintering in a quartz tube, however it remained unclear to me how the target has been shaped (it is supposed to be a disk?).

2. What is "ink" in TEM images in Suppl. materials?

3. Can Authors specify whether TEM cross section has been made on the same samples after performing all other studies? Or on analogous samples produced identically and with identical thickness?

4. Authors write "Superparamagnetism due to ferromagnetism" and "superparamagnetism most likely originates from the presence of a ferromagnetic order", which is not correct - SPM is actuallly a ferromagnetic state, it is not "induced" by ferromagnetism.

5. A small hump at 225 K and 265 K in Fig. S2 - did Authors verify their reproducibility, with enough resolution in temperature? Their current representation rather looks like a measurement artefact to me. Authors are also recommended to be more precise in their interpretation of the role of strained interfacial layer in this 740 nm thick film - it is almost a bulk sample! I highly doubt that any signatures of strained interface will be visible for such thickness.

6. What is the source and characteristics of reference "cubic MnSe powder" used in Fig. 2.?

7. Hysteresis loops in Fig. 3 shall be carefully verified. Authors report about magnetization of 20 nm (or 30 nm?) thick MnSe film being equal to 750 emu/cm3 which is huge and is about the magnetization value of a good ferromagnet like permalloy! Without more details on how accurate was the measurement and whether all possible ferromagnetic contamination has been excluded this data look doubtful to me. Comments about absence of coercivity in the hysteresis loop must be accompanied with the figure with zoomed-in region about zero field - it is not enough to show the magnetization curve in the field range of +/-2 T.

8. I suggest to provide more details on conclusion of epitaxial growth of films based on the analysis of the rocking curves as it is stated now in the present version of the manuscript.

9. Enhancement of susceptibility of 740 nm thick film (which can be safely considered as a bulk-like sample) as compared to powder MnSe is not discussed.

10. Finally, the hypothesis of superparamagnetism requires an additional prove/clarification due to following experimental facts which contradict this version:

- ZFC-FC curves do not show splitting and blocking temperature;

- Authors report a formation of epitaxial and continuos MnSe film excluding any Volmer-Weber (island-like) growth, consequently I do not see any reason why individual clusters might be formed;

- superparamagnetic behaviour occurs when anisotropy of magnetic particles become comparable to the thermal energy, thermal fluctuations lead to the stochastic flipping of magnetic moment of cluster. The mechanism why solid ferromagnetic film demonstrates superparamagnetic behaviour is unclear to me. Authors suspect the "superparamagnetism" coming from the strained interface, where tetragonal distortion unavoidably leads to an increase of magnetocrystalline anisotropy. First of all, this strained region also represents a continous film, not clusters. And second, this strain might play rather an opposite role - it will enhance anisotropy in tetragonally distorted film, in addition to induced ferromagnetism due to a charge transfer, as Authors claim. And enhanced anisotropy speaks against superparamagnetism.

Additional temperature-dependent hysteresis measurements in a broader temperature range and verification of susceptibility scaling as 1/T would be required to furhter discuss the origin of observed behaviour.

11. If Authors attribute the observed behaviour in resistivity and susceptibility to the strain from the substrate, a comparison of narrower thickness range would be beneficial. Interface contribution to the properties of 740 nm or even 140 nm thick film is negligible as the strain typically relaxes in a much thinner region near the substrate. To confirm the claimed conclusions would be beneficial to characterize the films with the intermediate thickness (20-50 nm and thinner), if such option exists.

12. Authors should provide more details on how the charge transfer is affected by the strain, whether it occurs also in non-strained (cubic) films, and what is the typical thickness of the film where these effects are expected to be prominent.

13. (related to Q3 above) Main text and Supplementary contain diverse thickness sets: 20/30, 140/130 and 750/760 nm - these are real 6 samples? If yes, the comparison of conductivity and magnetic properties for 20 and 30 nm sample would be of high interest as the role of interfacial effects such as strain and charge transfer is expected to be best resolvable for these two films.

Some minor issues:

used units are mixed (kbar, GPa, etc.) - it is advisable to unify them.

"valance", "we have check", polycrytalline (Fig. 2)

"when the (homogeneous) concentration of holes is higher than ." - sentence end is missing.

Comments on the Quality of English Language

Language and style is fully ok, final proof read is required to clear up few minor issues (see section above).

Reviewer 3 Report

Comments and Suggestions for Authors

Report on manuscript nanomaterials-2941860

Substrate charge transfer induced ferromagnetism in MnSe/SrTiO3 ultrathin films

by Chun-Hao Huang et al.

The authors report experiments on thin films of MnSe deposited on
strontiumtitanate. The films were characterized by TEM, XRD, XAS, and optical methods. The investigations included the temperature dependence of the resistivity and susceptibility as well as measurements of the magnetization.

In addition, the authors performed some ab-initio calculations on the electronic and magnetic structure. The results of the calculations are of limited validity because they are for bulk materials neglecting all interface and surface effects that play a major role the thinner the films become.
(This should be mentioned in the text as it is more important than the use of a
particular exchange- correlation functional.)

The investigations are comprehensive and the paper is well written, therefore, I recommend the manuscript to be published in Nanomaterials.
